# PINK1/Parkin-Mediated Mitophagy Partially Protects against Inorganic Arsenic-Induced Hepatic Macrophage Polarization in Acute Arsenic-Exposed Mice

**DOI:** 10.3390/molecules27248862

**Published:** 2022-12-13

**Authors:** Gaoyang Qu, Zi Liu, Jiaxin Zhang, Yaning Guo, Hui Li, Ruijie Qu, Wei Su, Huan Zhang, Lin Zhang, Hong Xu, Fuhai Shen, Shoufang Jiang, Heliang Liu, Jinlong Li

**Affiliations:** 1Hebei Key Laboratory for Organ Fibrosis Research, School of Public Health, North China University of Science and Technology, Tangshan 063210, China; 2Department of Internal Medicine Nursing, School of Nursing, Wannan Medical College, Wuhu 241000, China

**Keywords:** arsenic, liver, macrophage, mitophagy, PINK1, immunotoxicity

## Abstract

Inorganic arsenic is a well-known environmental toxicant and carcinogen, and there is overwhelming evidence for an association between this metalloid poisoning and hepatic diseases. However, the biological mechanism involved is not well characterized. In the present study, we probed how inorganic arsenic modulates the hepatic polarization of macrophages, as well as roles of PTEN-induced kinase 1 (PINK1)/Parkin-mediated mitophagy participates in regulating the metalloid-mediated macrophage polarization. Our results indicate that acute arsenic exposure induced macrophage polarization with up-regulated gene expression of inducible nitric oxide synthase (*Inos*) and arginase-1 (*Arg1*), monocyte chemotactic protein-1 (*Mcp-1*) and macrophage inflammatory protein-2 (*Mip-2*), tumor necrosis factor (*Tnf*)-α, interleukin (*Il*)-1β and *Il-6*, as well as anti-inflammatory factors *Il-4* and *Il-10*. In parallel, we demonstrated the disrupted hepatic redox balance typically characterized by the up-regulation of hydrogen peroxide (H_2_O_2_) and glutathione (GSH), and activation of PINK1/Parkin-mediated mitophagy in the livers of acute arsenic-exposed mice. In addition, our results demonstrate that it might be the PINK1/Parkin-mediated mitophagy that renders hepatic macrophage refractory to arsenic-induced up-regulation of the genes *Inos*, *Mcp-1*, *Mip-2*, *Tnf-α*, *Il-1β*, *Il-6* and *Il-4*. In this regard, this is the first time the protective effects of PINK1/Parkin-mediated mitophagy in inorganic arsenic-induced hepatic macrophage polarization in vivo have been reported. These findings add novel insights into the arsenical immunotoxicity and provide a basis for the preve.ntive and therapeutic potential of PINK1/Parkin-mediated mitophagy in arsenic poisoning.

## 1. Introduction

Inorganic arsenic is a ubiquitous element in nature [1]. Large quantities of epidemiological surveys and toxicological experiments have been devoted to clarify the relationship between arsenic-related diseases and metalloid poisoning surveys [2]. The latest research shows that inorganic arsenic has obvious immunotoxicity effects and can be used as an exogenetic stimuli to induce inflammatory responses and precancerous lesions of the liver, which is closely related to the formation of liver fibrosis and other diseases [3]. Inflammatory responses are thought to be started mainly by white blood cells such as neutrophils, mononuclear macrophages and natural killer (NK) cells. Compared to other organs, macrophages are abundant in the liver, at a rate of about 40 macrophages per 100 hepatocytes [4]. In recent years, it has been found that macrophages can be polarized into two different subsets of macrophages in different environments, in which classically activated macrophages have a strong ability to destroy microorganisms and tumor cells, and can secrete large numbers of proinflammatory cytokines such as tumor necrosis factor (TNF)-α, interleukin (IL)-1β and IL-6. Moreover, anti-inflammatory cytokines, such as IL-4 and IL-10, secreted by alternatively activated macrophages, play vital roles in the recovery of inflammation [5]. However, few studies have attempted to unravel the links between inorganic arsenic exposure and the polarization of macrophages in liver.

Oxidative stress as the main mechanism of arsenic poisoning has been widely recognized by scholars. Mitochondria dysfunction by excessive reactive oxygen species (ROS), of which more than 90% derived from oxidative phosphorylation on mitochondria inner membrane [6], could initiate inflammatory responses, which is of interest with regards to various diseases, including carcinogen [7]. Autophagy is a special mechanism for the non-selective removal of damaged cell components and harmful substances to maintain the stability of the intracellular environment [8], while mitophagy is a newly discovered cytoprotective mechanism that selectively scavenges dysfunctional mitochondria and thus antagonizes the outbreak of ROS-dependent inflammatory pathways [9]. Among various reported mitophagy, PTEN-induced kinase 1 (PINK1), a molecular receptor of mitochondrial damage, is particularly sensitive to depolarization of mitochondrial membrane potential (MMP) and can recruit and phosphorylate the E3 ubiquitin-protein ligase Parkin. Parkin then initiates ubiquitination voltage-dependent anion channel protein 1 (VDAC1) and other substrate proteins on the outer membrane of mitochondria. With the help of autophagy receptor sequestosome-1 (SQSTM1/p62), the ubiquitin-labeled mitochondria bind to microtubule-associated protein light chain 3 (LC3). Then, they are wrapped in double-layer autophagy vesicles to form mitochondrial autophagosomes, which are eventually fused with lysosomes and degraded by hydrolases [10]. Nevertheless, the method in which PINK1/Parkin-mediated mitophagy exerts itself in inorganic arsenic-mediated hepatic macrophage polarization remains largely unknown.

In this study, C57BL/6 mice were treated by a single oral administration of sodium arsenite (NaAsO_2_). Then, redox-related indexes, macrophage phenotypic molecules, chemokines and pro/anti-inflammatory cytokines were determined to probe hepatic redox status and macrophage polarization, respectively. Furthermore, we observed the effects of PINK1/parkin-mediated mitophagy upon arsenic exposure in the liver and attempted to confirm the effects of PINK1/Parkin-mediated mitophagy in arsenic-regulated macrophage polarization by restraining PINK1 expression with inhibitor ciclosporin A (CsA) in vivo. On the basis of intuitively understanding the hepatic immunotoxicity of inorganic arsenic, we attempt to provide new theoretical clues and an experimental basis for prevention and treatment of arsenic poisoning.

## 2. Results

### 2.1. Inorganic Arsenic Up-Regulates Expression of Hepatic Macrophage Phenotypic Molecules and Chemokines in Mice

The enzymes inducible nitric oxide synthase (iNOS) and arginase-1 (Arg1) are commonly used as specific phenotypic molecules for macrophage polarization [11]. In the present study, western blotting and real-time qPCR were performed to verify the effects of acute arsenic exposure on macrophage differentiation in the liver. As shown in Figure 1A,B, our data suggested that acute arsenic exposure time-dependently up-regulated protein expression of iNOS and Arg1 after 48 h. Simultaneously, treatment with the metalloid increased the gene expression of *Inos* after 48 h, while the transcription of gene *Arg1* started earlier, at 6 h. As key chemokines, MCP-1 and MIP-2 play vital roles in recruiting immune cells to pathologic sites, indirectly reflecting progress and regress of inflammation [12,13]. As shown in Figure 1C,D, our data showed that gene expression of *Mip-2* was drastically up-regulated upon acute arsenic exposure, and peaked at 6 h, which was 2.43 folds that of the control group. Consistent with gene expression of *Inos*, the metalloid also induced the gene expression of *Mcp-1*, which peaked at 48 h.

### 2.2. Inorganic Arsenic Induces Gene Expression of Hepatic Inflammatory Cytokines in Mice

Accompanied with up-regulated phenotypic molecules and chemokines, one additional peculiarity of macrophages is that they release a plethora of cytokines to communicate with other cells, thereby orchestrating immune responses. Next, we examined effects of the metalloid on gene expression of inflammatory cytokines by using real-time qPCR. As shown in Figure 2A–C, our data indicated that treatment with acute arsenic exposure time-dependently up-regulated transcription activity of the proinflammatory cytokines *Tnf-α* and *Il-β* in liver, which both peaked at 24 h (Figure 2A,B). Similarly, the metalloid induced the gene expression of *Il-6* at 6 h, then time-dependently decreased till 72 h (Figure 2C). After having confirmed up-regulation of proinflammatory cytokines, we next detected the expression of anti-inflammatory cytokines. Analogously, acute oral administration of arsenite also activated the transcription of genes *Il-4* and *Il-10*. In particular, the gene *Il-10* was 1.82 folds that of control group at 24 h. Together with all above results, we confirmed the hepatic macrophage polarization after a single oral administration of NaAsO_2_.

### 2.3. Inorganic Arsenic Induces Hepatic Redox Imbalance in Mice

In consideration of the proinflammatory property of ROS, it is then reasonable to speculate whether the metalloid can initiate ROS-dependent inflammatory responses in the liver. To judge the hepatic redox states upon arsenic exposure, we first determined the time course of redox-related indexes H_2_O_2_, GSH and MDA in the liver of mice treated by a single oral administration of NaAsO_2_. As shown in Figure 3A, we found that acute arsenic exposure time-dependently induced the accumulation of H_2_O_2_, a main type of ROS, in the liver [14]. H_2_O_2_ levels peaked at 6 h, which was 2.06 folds that of control group. GSH, the most effective antioxidant, existed in various tissues [15], increased in the liver at 24 h, and was higher than the control group by 87.38% (Figure 3B). MDA, an end production of lipid peroxidation [16], slightly increased at 6 and 24 h, which was not significant from control group (Figure 3C). These data together verified that acute arsenic exposure could induce hepatic redox imbalance in mice.

### 2.4. Inorganic Arsenic Activates PINK1/Parkin-Mediated Mitophagy in the Livers of Acute Arsenic-Exposed Mice

After having confirmed acute inorganic arsenic-induced hepatic redox imbalance and subsequent macrophage polarization in the liver, we ruminated about how to seek effective molecule targets to ameliorate arsenical oxidative and immune toxicity. Based on these considerations, we found that PINK1/Parkin-mediated mitophagy could effectively clear damaged mitochondria, preventing mtROS from bursting and provoking cascaded inflammation in the entire cells [17]. Consequently, western blotting and real-time qPCR were performed to detect hepatic protein expression of PINK1/Parkin-mediated mitophagy. As shown in Figure 4A–E, we observed the highest expression of PINK1 and Parkin at 24 h, which were 1.96 and 1.28 folds that of the control group, after the single oral administration of NaAsO_2_ (Figure 4A,D). Accompanied with the upregulation of PINK1 and Parkin, autophagy-related protein p62 and LC3 II/I were also verified to increase both at 6 and 72 h (Figure 4A,D). Consistently, mRNA levels of Pink1, Parkin, p62 and Lc3 were confirmed to increase in the liver at 6 or 24 h, after the treatment with the metalloid (Figure 4C,E). These data together suggested that acute arsenic exposure activated the PINK1/Parkin-mediated mitophagy in liver.

### 2.5. PINK1/Parkin-Mediated Mitophagy Resists Arsenic-Induced Gene Expression of Inflammatory Cytokines in Liver

To further probe how PINK1/Parkin-mediated mitophagy exerts in inorganic arsenic-mediated hepatic macrophages polarization, we tried to constrain PINK1/Parkin-mediated mitophagy by inhibiting PINK1 expression with the pharmacological inhibitor CsA. As shown in Figure 5A, intraperitoneal injection with 10 mg/kg CsA significantly suppressed the inorganic arsenic-induced PINK1 expression and demonstrated that the pharmacological inhibition of PINK1 appeared to be conducted successfully in the livers of arsenic-exposed mice. In addition, the pharmacological inhibition of PINK1 promoted the inorganic arsenic-induced production of H_2_O_2_, which represented the anabatic redox imbalance (Figure 5B). Driven by the hypothesis that PINK1/Parkin-mediated mitophagy could resist arsenic-induced up-regulation of phenotypic molecules, chemokines and pro/anti-inflammatory cytokines, we screened above targets potentially regulated by PINK1/Parkin-mediated mitophagy. As shown in Figure 5C–G, our results denoted that restrained PINK1/Parkin-mediated mitophagy by CsA could promote arsenic-induced gene expression of *Inos*, *Mcp-1*, *Mip-2*, *Tnf-α*, *Il-1β*, *Il-6 and Il-4*, while analogous effects were not observed for *Arg1* and *Il-10* in liver (Figure 5D,K). Therefore, these data jointly suggested that PINK1/Parkin-mediated mitophagy might suppress the expression of *Inos*, *Mcp-1*, *Mip-2*, *Tnf-α*, *Il-1β*, *Il-6 and Il-4*, which partially protects against arsenic-induced macrophage polarization in liver.

## 3. Discussion

More recent research has reported that the liver is not only a metabolic and detoxifying organ, but also an important immunological organ with numerous innate and adaptive immune cells [18]. Mitophagy has been proved to sweep damaged mitochondria away in a timely manner to protect cellular environmental homeostasis from oxidative imbalance and subsequent tissue inflammation. However, the mechanism in which PINK1/Parkin-mediated mitophagy participates in regulating inorganic arsenic-modulated hepatic macrophage polarization have not been elucidated. In the current study, we demonstrated that acute arsenic exposure could induce hepatic macrophage polarization with up-regulation of phenotypic molecules, chemokines and inflammatory cytokines. Besides these, PINK1/Parkin-mediated mitophagy might inhibit arsenic-induced gene expression of phenotypic molecules, chemokines and inflammatory cytokines, which partially restrain macrophage polarization in liver.

Macrophages play vital roles in immune defense and provide them with innate immune surveillance in the liver, in which macrophages are the largest group of innate immune cells [19]. Generally speaking, classically activated macrophages express high levels of iNOS that compete with Arg1 for l-arginine. By inhibiting iNOS, Arg1 may promote the alternatively activated macrophages and contributes to the suppression of the classically activated macrophages [20,21]. Judging from the up-regulation of iNOS and Arg1, our results revealed that a single oral administration of NaAsO_2_ obviously initiated the pro/anti-inflammatory macrophage differentiation, eliciting inflammatory responses. Analogously, phenotypic molecules iNOS and Arg1 were also noted to increase in the livers of mice exposed to arsenite [22,23]. Compelling evidence exists that show that macrophage infiltration into the liver is primarily controlled by the CeC chemokine receptor 2 (CCR2) and its main ligand MCP-1 in mice [12]. In addition, MIP-2, a classical chemokine of neutrophil, could also be secreted by activated macrophages in the early stage of inflammation to mobilize other immune cells, exacerbating the inflammatory responses. In this manuscript, we found that acute arsenic treatment persistently induced expression of gene *Mcp-1* and *Mip-2*, which also were observed in arsenic-induced liver fibrosis in mice [24].

Macrophages promote inflammatory cascades by secreting various cytokines, the signaling proteins that are produced transiently and exert pleiotropic effects on cells, to initiate and constrain inflammatory responses to pathogens and injury. Among these cytokines, TNF-α is a potent proinflammatory cytokine secreted by immune cells, particularly activated macrophages, but also neutrophils, dendritic cells and NK cells [25]. Normally, IL-6 contributes to host defense through the stimulation of acute phase responses. Once dysregulated, continual synthesis of IL-6 will play a pathological effect on chronic inflammation [26]. IL-1β is one of the most crucial mediators of inflammation and host responses to pathological damage [27]. In the current research, we verified hepatic inflammation stimulated by the external metalloid with up-regulation of proinflammatory cytokines TNF-α, IL-1β and IL-6. In agreement with our results, it has been reported that TNF-α, IL-1β and IL-6 were dramatically elevated in the liver of arsenic-treated mice or rats [24,28,29]. Not only that, macrophages also secrete a cascade of other anti-inflammatory cytokines such as IL-4 and IL-10. IL-10, produced mainly by alternatively activated macrophages, T helper 2 cells and regulatory T cells, is an important immuno-regulatory cytokine [30]. This series of anti-inflammatory cytokines limits and terminates inflammatory responses by inhibiting the synthesis of many pro-inflammatory cytokines and regulating the differentiation and proliferation of macrophages. Previous studies revealed that sustained arsenic exposure increased the IL-4 and IL-10 levels in serum and liver, respectively [31,32], which is concordant with our findings of enhanced gene expression of *Il-4* and *Il-10* in the livers of acute arsenic-exposed mice.

More than 90% of ROS is derived from mitochondria, and excessive ROS could accelerate mitochondria dysfunction and diffuse to the cytosol, acting as a signaling molecule to trigger a pathological reaction, which generates a undesirable feedback loop [33]. To confront redundant accumulation of H_2_O_2_, we speculated that reductive GSH was promptly produced and effectively ameliorated H_2_O_2_ accumulation in liver, withstanding the subsequent production of MDA. Concordant with our results, previous research reported that treatment of 5–20 mg/kg NaAsO_2_ induced elevation of MDA and depletion in physiological antioxidant content such as superoxide dismutase (SOD) and catalase (CAT) in the liver [34,35,36]. Overall, our results therefore jointly revealed the obvious redox imbalance upon acute arsenic exposure. Mitophagy is a vital form of autophagy for the selective removal of dysfunctional or redundant mitochondria. Accumulating evidence implicate the elimination of dysfunctional mitochondria as a powerful means employed by autophagy to keep the redox state and immune system in check [37,38]. PINK1/Parkin is the best characterized signaling pathway so far and is recognized as the major regulatory system involved in mitophagy [39]. In our current study, we found that compromised PINK1/Parkin-mediated mitophagy could promote the inorganic arsenic-induced production of H_2_O_2_, as well as gene expression of *Inos*, *Mcp-1*, *Mip-2*, *Tnf-α*, *Il-1β*, *Il-6* and *Il-4*, which demonstrates that PINK1/Parkin-mediated mitophagy might ameliorate production of ROS, partially protecting against arsenic-induced macrophage polarization in liver. Consistent with this finding, Patoli et al. reported that the inhibition of mitophagy was an early feature of macrophage activation, which efficiently promoted an increase in macrophage activation markers including CD64, CD80, TNF-α, IL-6 and iNOS. In the absence of autophagy, release of mitochondrial DNA (mtDNA), a damage-associated molecular patterns (DAMPs), was verified to enhanced production of IL-6 through the activation of the nuclear factor kappa B subunit (NF-κB) pathway via the toll like receptor 9 (TLR9) [40]. Moreover, Xu et al. observed that silencing of PINK1 amplified mtDNA-NLRP3 association in the presence of anoxia/reoxygenation (A/R), as well as release of IL-1β and overexpression of PINK1 diminished above the effects [38]. Sliter et al. subjected wild-type, *Pink1*^−/−^, and *Parkin*^−/−^ mice to exhaustive exercise, then observed that multiple cytokines such as IL-6, IL-12, IL-13, MCP-1 and MIP-1β were elevated in the serum of *Pink1*^−/−^ or *Parkin*^−/−^ mice [41].

In this paper, our results are partially in contrast to previous reports of investigations with experimental animals, such as preponderant alternatively activated macrophage polarization with permanent arsenic exposure [23]. The variations may be related to different exposure durations, doses, and/or experimental systems. We speculated that the both activation of classically activated and alternatively activated macrophages was due to a temporary immune response upon a single oral administration of arsenite. Sustained arsenic exposure studies as well as effects of PINK1/Parkin-mediated mitophagy involved were thus extremely necessary. At present, many puzzles concerning the role of mitophagy in the immune system and disease context remain to be solved. Not only that, research into the crosstalk of distinctive forms of mitophagy in the regulation of immunity is still lacking. Based on these above, more experiments are required to demonstrate the potential mechanism, by which PINK1/Parkin-mediated mitophagy effectively restricts inorganic arsenic-induced immune injury in vivo.

## 4. Materials and Methods

### 4.1. Animals and Experimental Procedures

Thirty-six female C57BL/6 mice (weighing 18–23 g, 6–8 weeks old) were purchased from Beijing Huafukang Biotechnology Co., Ltd. [SCXK (Jing)2019-0008; Beijing, China]. Upon arrival, the mice were maintained in a 12-h/12-h light/dark cycle and provided with food and water ad libitum for 1 week before the experiment. All experiments and surgical procedures were approved by the Committee on the Ethics of North China University of Science and Technology (LX2019033), which complies with the National Institutes of Health Guide for the Care and Use of Laboratory Animals. All efforts were made to minimize the number of animals used and their suffering.

The sodium arsenite (NaAsO_2_, ≥99.0%) was purchased from Sigma Chemical Co. (St. Louis, MO, USA). The concentration of NaAsO_2_ was selected on the basis of previously published studies [42,43] as well as our preliminary experiments (10% LD50). Acute arsenic exposure murine model was conducted by NaAsO_2_ (10 mg/kg) intragastrically for 6, 24, 48 and 72 h, respectively. In addition, pretreatment with a single intraperitoneal administration of ciclosporin A (CsA, 10 mg/kg) for 2 h, then mice were exposed to NaAsO_2_ (10 mg/kg) intragastrically for 6/24 h. Control mice were treated with saline only. At each end point of the treatment, all mice were weighed and killed by ether anesthesia. The entire liver of the control and acute arsenite-exposed mice were promptly removed and weighed, and then stored at −80 °C for future use.

### 4.2. Western Blot Analysis

Western blotting was performed as previously described [43]. Primary antibodies included polyclonal rabbit antibodies to PINK1 (A7131, Company ABclonal, Wuhan, China), Parkin (ET1702-60, Hangzhou Huaan Biotechnology Co., Ltd. Hangzhou, China), SQSTM1/p62 (PM045, MBL Beijing Biotech Co., Ltd. Beijing, China), LC3 (PM036, MBL Beijing Biotech Co., Ltd. Beijing, China), and β-actin (66009-1-IG, Proteintech Group, Inc., Rosemont, IL, USA) at a dilution of 1:1000. After three washes and incubation with goat anti-rabbit or anti-mouse secondary antibodies (S1001/S1002, SeraCare, Gaithersburg, MD, USA) at a dilution of 1:5000 in blocking buffer, immunoblots were visualized using ECL prime Western blotting detection reagent (ZD310A, ZomanBio, Beijing, China). The results were normalized against the β-actin expression level and corresponding control.

### 4.3. Total RNA Isolation and Real-Time qPCR Analysis

Total RNA of the liver from experimental mice was isolated using a Trizol Reagent (Invitrogen, Carlsbad, CA, USA). Real-time PCR was conducted using a twostep method with an ABI QuantStudio™ 6 Flex PCR System (Applied Biosystems, Inc., Norwalk, CT, USA). Briefly, 500 ng of total RNA was reverse-transcribed to cDNA using M5 Super plus qPCR RT kit with gDNA remover (Mei5 Biotechnology, Co., Ltd., Beijing, China), and PCR amplification was performed by 2X M5 HiPer Realtime PCR Super mix with Low Rox kit (Mei5 Biotechnology, Co., Ltd., Beijing, China). PCR amplification conditions were: 1 cycle of initial denaturation (95 °C for 30 s), and 40 cycles of amplification (95 °C for 5 s and 60 °C for 34 s). Primers for mouse genes designed by PRIMER 3 software and synthesized by RuiBiotech (Beijing, China) were shown in Table 1.

### 4.4. Analysis of H_2_O_2_ Content in Liver

The liver was isolated and washed with normal saline to remove blood and clots. Then, homogenate was centrifuged and the supernatant was used for biochemical analyses. The protein concentration in the supernatant was determined by the BCA Protein Assay Kit (EpiZyme, Shanghai, China). The content of H_2_O_2_ in gastric submucosal arteries was assessed using a commercially available kit (Jiancheng Biological Institute, Nanjing, China). H_2_O_2_ was bound with molybdenic acid to form a complex, which was measured at 405 nm and the content of H_2_O_2_ was then calculated. The levels of H_2_O_2_ in liver were finally expressed as mmol/g protein.

### 4.5. Analysis of GSH Levels in Liver

GSH levels were determined by the modified 5,5′-dithiobis 2-nitrobenzoic acid (DTNB) method using a commercially available kit according to the manufacturer’s recommended protocol (Jiancheng Biological Institute, Nanjing, China). Briefly, the liver from mice was washed with normal saline to remove blood and clots, and then homogenized on ice with 5 mL 5% trichloroacetic acid (TCA) per gram of tissue weight. Homogenates were centrifuged at 1000× *g* for 15 min at 4 °C and the aliquots samples of the supernatants were then used for the analysis of GSH. The levels of GSH in liver were finally expressed as μmol/g protein.

### 4.6. Analysis of MDA Levels in Liver

The livers of experimental mice were homogenized on ice with 9 mL (5 mmol/L containing 2 mmol/L EDTA, PH 7.4) per gram of tissue weight. Homogenates were then centrifuged at 1000× *g* for 10 min at 4 °C and the supernatants were used for the analysis of MDA in liver according to each manufacturer’s recommended protocol (Jiancheng Biological Institute, Nanjing, China). The thiobarbituric acid reaction (TBAR) method was used to determine MDA and the levels were expressed as nmol/mg protein.

### 4.7. Statistical Analysis

A statistician was consulted before the start of the experiment for the minimum number of mice required to give viable statistical and reproducible data and for statistical analysis. Data were presented as mean ± SD. All statistical analyses were performed using Graphpad Prism 8 software (GraphPad Software, San Diego, CA, USA). One-way ANOVA with Dunnett-*t* or independent-Samples *t*-test were performed, depending on the data.

## 5. Conclusions

In summary, our results indicate that acute arsenic exposure disrupted the hepatic redox balance and induces macrophage polarization by up-regulating expression of characteristic phenotypic molecules, chemokines and pro/anti-inflammatory cytokines. Not only that, it is possible that the PINK1/Parkin-mediated mitophagy renders hepatic macrophage refractory to arsenic-induced up-regulation of gene *Inos*, *Mcp-1*, *Mip-2*, *Tnf-α*, *Il-1β*, *Il-6* and *Il-4*. In this regard, this is the first time that the protective effects of PINK1/Parkin-mediated mitophagy in inorganic arsenic-induced hepatic macrophage polarization in vivo have been reported. Based on these experimental data, it appears that this research further replenishes arsenical immunotoxicity and provides new theoretical clues to protect against arsenic poisoning.

## Figures and Tables

**Figure 1 molecules-27-08862-f001:**
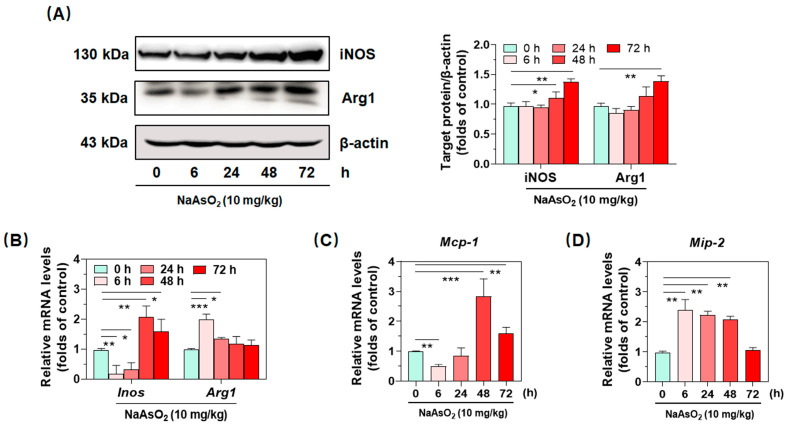
Up-regulated expression of macrophage phenotypic molecules and chemokines in the livers of acute arsenic-exposed mice. (**A**) hepatic protein expression of macrophage-specific phenotypic molecule iNOS and Arg1 were determined by western blotting, of which protein densities are shown, gene expression of *Inos* and *Arg1* (**B**), as well as chemokine *Mip-2* and *Mcp-1* (**C**,**D**) were determined by real-time qPCR, of which relative mRNA levels normalized to *Gapdh* are shown at different time points after an oral administration of NaAsO_2_. Data are presented as mean ± SD (*n* = 4). * denotes *p* < 0.05 compared with the control group. ** denoted *p* < 0.01 compared with the control group. *** denotes *p* < 0.001 compared with the control group.

**Figure 2 molecules-27-08862-f002:**
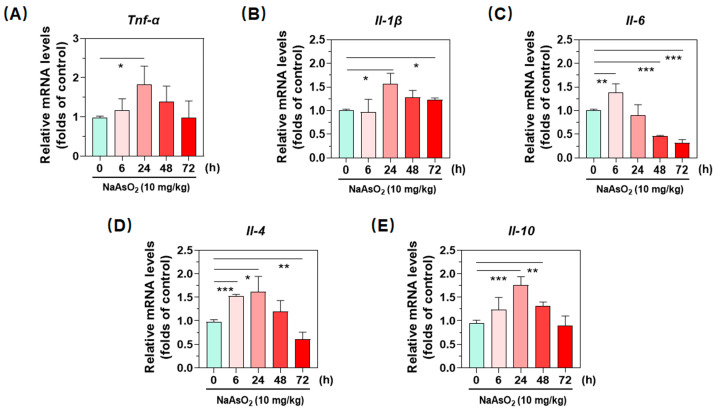
Enhanced gene expression of inflammatory cytokines in the livers of acute arsenic-exposed mice. (**A**–**C**) gene expression of proinflammatory cytokine *Tnf-α*, *Il-1β* and *Il-6*, (**D**,**E**) as well as anti-inflammatory cytokines *Il-4* and *Il-10* in liver were determined by real-time qPCR, of which relative mRNA levels normalized to *Gapdh* are shown at different time points after an oral administration of NaAsO_2_. Data are presented as mean ± SD (*n* = 4). * denotes *p* < 0.05 compared with the control group. ** denotes *p* < 0.01 compared with the control group. *** denotes *p* < 0.001 compared with the control group.

**Figure 3 molecules-27-08862-f003:**
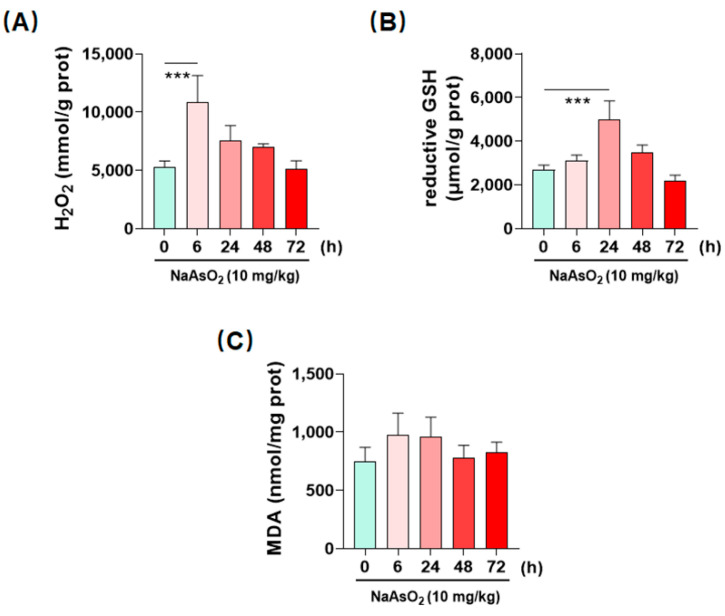
Imbalanced redox homeostasis in the liver of acute arsenic-exposed mice. (**A**–**C**) redox-related indexes H_2_O_2_, GSH and MDA in the liver were detected by biochemical kits at different time points after an oral administration of NaAsO_2_. Data are presented as mean ± SD (*n* = 4). *** denotes *p* < 0.001 compared with Control (0 h) group.

**Figure 4 molecules-27-08862-f004:**
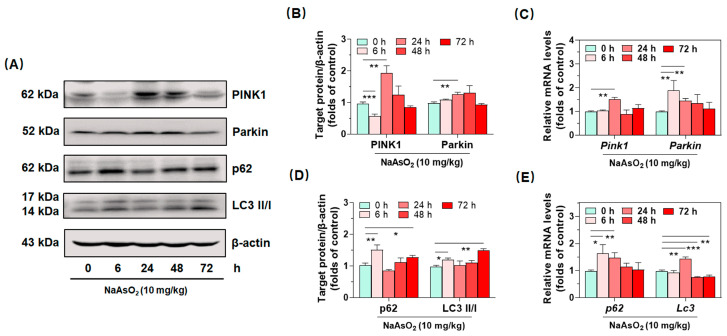
Activated PINK1/Parkin-mediated mitophagy in the livers of acute arsenic-exposed mice. (**A**) hepatic expression of proteins PINK1, Parkin, p62, LC3II/I were determined by Western blotting, of which protein densities are shown (**B**,**D**), (**C**,**E**) gene expression of *Pink1*, *Parkin*, *p62* and *Lc3* were determined by real-time qPCR, of which relative mRNA levels normalized to *Gapdh* are shown, at different time points after an oral administration of NaAsO_2_. Data were presented as mean ± SD (*n* = 4). * denotes *p* < 0.05 compared with the control group. ** denotes *p* < 0.01 compared with the control group. *** denotes *p* < 0.001 compared with the control group.

**Figure 5 molecules-27-08862-f005:**
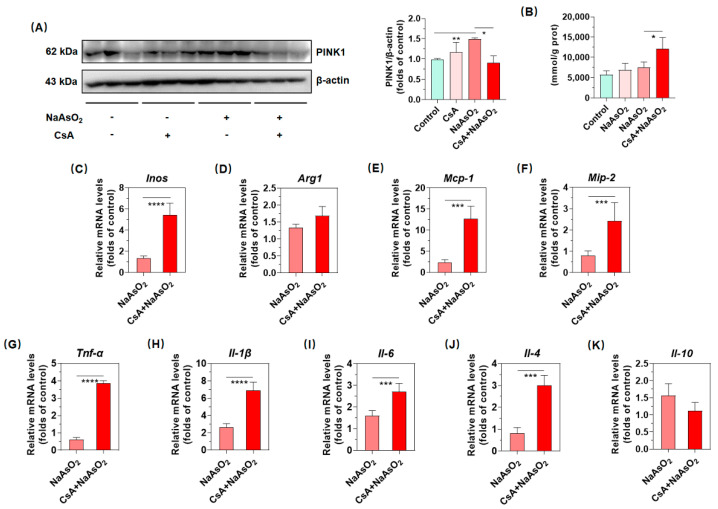
Compromised PINK1/Parkin-mediated mitophagy strengthened arsenic-induced gene expression of inflammatory cytokines in liver. (**A**) hepatic expression of protein PINK1 were determined by Western blotting, of which protein densities are shown (**B**), (**C**–**K**) gene expression of *Inos*, *Arg1*, *Mcp-1*, *Mip-2*, *Tnf-α*, *Il-1β*, *Il-6, Il-4* and *Il-10* were determined by real-time qPCR, of which relative mRNA levels normalized to *Gapdh* in the livers of arsenic-exposed mice after pretreatment with 10 mg/kg CsA for 2 h are shown. Data are presented as mean ± SD (*n* = 6). * denotes *p* < 0.05 compared with the NaAsO_2_ group. ** denotes *p* < 0.01 compared with the control group. *** denoted *p* < 0.001 compared with the NaAsO_2_ group. **** denotes *p* < 0.001 compared with the NaAsO_2_ group.

**Table 1 molecules-27-08862-t001:** Primer Sequences Used for the Amplification of Each Gene.

Gene Name	Primer Sequence (5′–3′)	Amplicon Size (bp)
Accession Number		
*iNOS*	ACCCCTGTGTTCCACCAGGAGATGTTGAA	189
(NM_001313922.1)	TGAAGCCATGACCTTTCGCATTAGCATGG	
*Arg1*	CTCCAAGCCAAAGTCCTTAGAG	185
(NM_007482.3)	AGGAGCTGTCATTAGGGACATC	
*Mcp-1*	TGAGTAGGCTGGAGAGCTACAA	123
(NM_053647.1)	ATGTCTGGACCCATTCCTTC	
*Mip-2*	CCCCAAAGGGATGAGAAGTTC	323
(NM_053647.1)	GGCTTGTCACTCGAATTTTGAGA	
*Tnf-α*	CCCCAAAGGGATGAGAAGTTC	101
(NM_013693)	GGCTTGTCACTCGAATTTTGAGA	
*Il-1β*	TGACCTGGGCTGTCCTGATG	160
(NM_008361)	GGTGCTCATGTCCTCATCCTG	
*Il-6*	CTGCAAGAGACTTCCATCCAG	131
(NM_031168)	AGTGGTATAGACAGGTCTGTTGG	
*Il-4*	GGTCTCAACCCCCAGCTAGT	102
(NM_021283)	GCCGATGATCTCTCTCAAGTGAT	
*Il-10*	GGGGCCAGTACAGCCGGGAA	101
(NM_010548)	CTGGCTGAAGGCAGTCCGCA	
*Pink1*	CACACTGTTCCTCGTTATGAAGA	157
(NM_036164400.1)	CTTGAGATCCCGATGGGCAAT	
*Parkin*	TCTTCCAGTGTAACCACCGTC	115
(NM_NM_016694.4)	GGCAGGGAGTAGCCAAGTT	
*Sqstm1/p62*	GAACTCGCTATAAGTGCAGTGT	131
(NM_001290769.1)	AGAGAAGCTATCAGAGAGGTGG	
*Map1lc3b*	CGCTTGCAGCTCAATGCTAAC	93
(NM_001364358.1)	CTCGTACACTTCGGAGATGGG	
*Gapdh*	TGTGTCCGTCGTGGATCTGA	150
(NM_001289726.1)	TTGCTGTTGAAGTCGCAGGAG	

## Data Availability

All datasets generated in this study are included in the article.

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
