# Peer review of "PINK1/Parkin-Mediated Mitophagy Partially Protects against Inorganic Arsenic-Induced Hepatic Macrophage Polarization in Acute Arsenic-Exposed Mice"

_molecules, 2022, doi:10.3390/molecules27248862_

Round 1
Reviewer 1 Report
The manuscript reports an investigation of the inflammatory effects in the liver of mice caused by acute arsenic exposure. In particular, the induction of M1/M2 macrophage polarization was observed through the western blotting and real-time qPCR measurements of phenotypic molecules, chemokines, and pro/anti-inflammatory cytokines. Moreover, the protective effects of PINK1/Parkin-mediated mitophagy in inorganic arsenic-induced hepatic macrophage polarization was reported for the first time. The manuscript is clearly presented, the methodology is reliable, the results are adequately stated, and the discussion is reasonable. Only a few typos/wordings need to be corrected, such as:
(1) In line 27, the numbers in “H2O2” should be in subscripts.
(2) In line 58, does “inter membrane” means “inner membrane”, “outer membrane”, or “intermembrane space”? There is no such thing as “inter membrane” and “inter” is not a word but just a word root.
(3) In line 221, same as (a), it is not clear what “inter environment” means.
Reviewer 2 Report
Interesting work is presented by the authors. A small number of questions have arisen. It is unclear why the authors interpret the data obtained within the M1M2 paradigm. It is worth reading, for example, the works of Murray https://pubmed.ncbi.nlm.nih.gov/25035950/ , which presents the concept of phenotypic continuum, or more radical views that suggest abandoning the M1V2 paradigm altogether because of the oversimplified view of the functional plasticity of macrophages, as well as the lack of clear M1 and M2 phenotypes (https://www.ahajournals.org/doi/10.1161/CIRCRESAHA.116.309194?url_ver=Z39.88-2003&rfr_id=ori:rid:crossref.org&rfr_dat=cr_pub%20%200pubmed
https://www.cell.com/immunity/fulltext/S1074-7613(20)30230-2?_returnURL=https%3A%2F%2Flinkinghub.elsevier.com%2Fretrieve%2Fpii%2FS1074761320302302%3Fshowall%3Dtrue). Another comment concerns the methods of statistical processing; the authors cite methods used only for the normal distribution of traits. Has this always been observed, what tests of the normal distribution were applied?
Round 2
Reviewer 2 Report
All questions were answered satisfactorily.